# Tissue Doppler Imaging Provides Incremental Value in Predicting Six Months In-Stent Restenosis in Patients with Coronary Artery Disease

**DOI:** 10.3390/diagnostics15050579

**Published:** 2025-02-27

**Authors:** Jih-Kai Yeh, Victor Chien-Chia Wu, Fen-Chiung Lin, I-Chang Hsieh, Po-Cheng Chang, Chun-Chi Chen, Chia-Hung Yang, Wen-Pin Chen, Kuo-Chun Hung

**Affiliations:** 1Department of Cardiology, Chang Gung Memorial Hospital, Linkou Medical Center, Taoyuan City 33305, Taiwan; 2College of Medicine, Chang Gung University, Taoyuan City 33302, Taiwan; 3Graduate Institute of Clinical Medicine, Taipei Medical University, Taipei City 11031, Taiwan; 4Graduate Institute of Pharmacology, National Taiwan University, Taipei City 10617, Taiwan

**Keywords:** Tissue Doppler Imaging, in-stent restenosis, coronary artery disease

## Abstract

**Background:** Invasive coronary angiography is the gold standard for assessing in-stent restenosis (ISR) in patients with coronary artery disease. However, the predictive value of non-invasive Tissue Doppler Imaging (TDI) to evaluate patients with ISR has not been studied extensively. **Methods:** A total of 41 patients (19 with acute myocardial infarction and 22 with stable angina pectoris) who received percutaneous coronary intervention (PCI) were enrolled in the study. Time-to-peak velocities (TpV) of 12 non-apical segments of the left ventricle, by pulse wave TDI echocardiography, were obtained within two days prior to the PCI and six months later. **Results:** A 12-segmental mean TpV ≥ 279.6 ms at six months after PCI was able to detect ISR (odds ratio: 2.09, 95% CI 1.004–4.352, *p* = 0.049). Moreover, a significant decrease in the standard deviation of TpV was demonstrated in patients without ISR (85.8 ± 44.8 vs. 60.3 ± 31.7 ms, *p* = 0.001), but not in patients with ISR (97.7 ± 53.3 vs. 91.2 ± 52.6 ms, *p* = 0.57). **Conclusions:** Pulse-wave TDI echocardiography is a promising tool in the detection of ISR six months after PCI in patients with coronary artery disease.

## 1. Introduction

Repeated coronary angiography has been the gold standard for assessing in-stent restenosis (ISR) in patients with ischemia symptoms after coronary intervention. Even with advancements in non-invasive imaging of computed tomography, its diagnostic utility remains questionable due to blooming artifacts for metallic stent strut [1]. Tissue Doppler Imaging (TDI) has emerged as a potential non-invasive tool in the diagnosis of coronary artery disease with subtle regional and global left ventricular (LV) myocardial wall motion abnormalities over the past 10 years [2,3]. Ischemic myocardial tissue creates areas of slow conduction, and TDI has been shown to be an excellent modality for investigating timing of velocity measurement related to regional function and the resultant electrical and mechanical dispersion [4,5]. Pulse-wave Tissue Doppler Imaging (PWTDI) study has the advantages of range-gating and a high frame rate (150–200 frames/second), which enable the detection of peak myocardium contraction from a specific region of interest (ROI) with high spatial (~1 mm) and temporal (~5 ms) resolution, beyond visual capabilities (60–80 ms).

Previous studies have reported that left ventricular ejection fraction (LVEF), wall motion score index, conventional Doppler LV filling flow pattern index, myocardial performance index, speckle tracking echocardiography, and stress echocardiography provide valuable prognostic information in patients after acute myocardial infarction (AMI). LV myocardial ischemia-induced dyssynchrony indices have also been identified as independent predictors of mortality in patients after AMI [6,7]. Tissue Time-to-Peak Velocity (TpV) is an echocardiographic parameter derived from PW-TDI that reflects the time interval from the onset of myocardial motion to the peak velocity of myocardial contraction. It provides insight into myocardial contractile timing, mechanics, and synchronization, which are crucial determinants of LV function and performance. However, the potential of TDI-derived TpV as an indicator of myocardial ischemia-related longitudinal dyssynchrony in predicting ISR has not been fully investigated. This study aimed to determine whether LV longitudinal myocardial contraction, assessed using PWTDI, could serve as a predictive marker for ISR in patients with coronary artery disease (CAD) undergoing percutaneous coronary intervention (PCI), evaluated six months post-procedure.

## 2. Materials and Methods

### 2.1. Patient Enrollment and Baseline Characteristics

Patients were prospectively enrolled in the study based on clinical indications for coronary angiography (CAG). These included individuals presenting with symptomatic angina pectoris accompanied by objective evidence of myocardial ischemia, as well as those with documented AMI who required PCI. Additionally, all patients with a history of prior PCI and stent placement underwent follow-up assessments as part of routine post-PCI care. In these cases, CAG was performed to evaluate stent patency and guide further management. This study design allowed for both pre- and post-procedure echocardiographic assessments to capture longitudinal changes in myocardial function.

Upon admission, all patients were thoroughly evaluated for eligibility. Inclusion criteria required patients to be capable of providing informed consent and completing both baseline and follow-up visits, including repeat echocardiography when necessary. Exclusion criteria included severe non-cardiac conditions, such as terminal malignancies, which might compromise follow-up compliance or affect study outcomes. A total of 41 patients were ultimately enrolled, including 19 with AMI and 22 with stable angina pectoris. The study protocol was approved by the Institutional Review Board (IRB) of Chang Gung Medical Foundation (Approval No.: [201901586B0], Date: 18 May 2019), and all procedures adhered to the ethical principles outlined in the Declaration of Helsinki.

Baseline characteristics were meticulously documented at enrollment to account for potential confounding factors. Key clinical data included demographic information (age, sex), cardiovascular risk factors (diabetes mellitus, hypertension, hyperlipidemia, and smoking status), and prior cardiac interventions. Detailed records of stent location (e.g., left anterior descending artery [LAD], left circumflex artery [LCX], and right coronary artery [RCA]) and the number of stented vessels were maintained. Concurrent medications such as aspirin, beta-blockers, angiotensin-converting enzyme inhibitors (ACEIs), angiotensin II receptor blockers (ARBs), and calcium channel blockers (CCBs) were also recorded.

### 2.2. Echocardiographic Assessment and Tissue Doppler Imaging

After obtaining informed consent, patients underwent baseline echocardiographic evaluation, including PWTDI, within two days prior to their scheduled CAG and PCI. The objective of this baseline assessment was to evaluate myocardial contractility and dyssynchrony using TpV and peak velocity (Vp) parameters derived from Doppler measurements.

All echocardiographic studies were performed using a commercially available Vivid 7 echocardiography system (1.7/3.5 MHz transducer, GE-Vingmed Ultrasound, Horten, Norway). Following established protocols, a sample volume of 6 mm was positioned at the mid-myocardium across 12 non-apical segments of the LV. These segments were visualized through three standard apical views. From the apical four-chamber (A4C) view, mid-septal (MS), baso-septal (BS), mid-lateral (ML), and baso-lateral (BL) segments were assessed. In the apical two-chamber (A2C) view, measurements were obtained from the mid-anterior (MA), baso-anterior (BA), mid-inferior (MI), and baso-inferior (BI) segments. Finally, the apical three-chamber (A3C) view provided access to the mid-anteroseptal (MAS), baso-anteroseptal (BAS), mid-posterior (MP), and baso-posterior (BP) segments.

TpV was measured as the interval from the electrocardiographic R-wave to the point of peak myocardial velocity directed toward the apex (Figure 1). Each measurement was repeated for accuracy, and the average TpV across all 12 segments was calculated to provide a comprehensive index of myocardial contractile function. Additionally, the standard deviation of TpV (SDTpV) was recorded as a measure of regional dyssynchrony (Figure 2). These parameters have been validated in previous research as indicators of myocardial performance and ischemic dysfunction [6].

### 2.3. Follow-Up Procedures and In-Stent Restenosis Evaluation

Patients were scheduled for follow-up visits approximately six months after PCI. At these visits, repeated echocardiographic assessments were conducted to evaluate longitudinal changes in TpV, Vp, and other functional parameters. In cases where clinical symptoms suggested possible ISR, follow-up coronary angiography was performed to confirm the diagnosis. ISR was defined as ≥50% luminal diameter stenosis within previously stented coronary segments, as visualized on angiography. The findings were correlated with echocardiographic parameters to assess the predictive value of TpV. LVEF was measured using biplane contrast ventriculography during cardiac catheterization. Changes in LVEF, alongside TpV and SDTpV, were analyzed to explore their association with ISR and other cardiac events. Patients without evidence of ISR served as a control group for comparative analyses.

### 2.4. Statistical Analysis

Continuous variables were expressed as mean ± standard deviation (SD) or median with interquartile range (IQR), depending on the data distribution. Categorical variables were presented as frequencies and percentages. The Shapiro-Wilk test was used to assess the normality of continuous data.

For normally distributed data, Student’s *t*-tests were used to compare differences between independent groups, and paired *t*-tests were applied to evaluate changes in echocardiographic parameters between baseline and follow-up. In cases where non-normality was detected, appropriate non-parametric tests were implemented, including the Mann-Whitney U test for comparisons between independent groups and the Wilcoxon signed-rank test for paired data. Differences in categorical variables were analyzed using chi-square test or Fisher’s exact test, as appropriate.

To identify independent predictors of ISR, forward stepwise logistic regression was performed. TpV and other Doppler parameters were included as candidate variables. The predictive performance of TpV was further evaluated using receiver operating characteristic (ROC) curve analysis, which provided optimal cut-off values, sensitivity, and specificity for detecting ISR. The reproducibility of TpV and Vp measurements was assessed through inter- and intra-observer variability studies, yielding coefficients of variation of 4% (r = 0.92) and 2.5% (r = 0.95), respectively. All statistical analyses were conducted using SPSS software (Version 19.0, SPSS Inc., Chicago, IL, USA). A two-tailed *p*-value of <0.05 was considered statistically significant for all tests.

## 3. Results

### 3.1. Patient Characteristics: Angina vs. AMI or ISR vs. Non-ISR

The baseline characteristics of patients, categorized by their primary condition (angina pectoris or AMI) are summarized in Table 1. No significant differences were observed between the two groups in terms of demographic and clinical variables, with the exception of a higher smoking rate (52.6% vs. 18.2%, *p* = 0.026) and higher diastolic blood pressure (76.9 ± 13.7 mmHg vs. 69.5 ± 8.0 mmHg, *p* = 0.038) in the AMI group. Both groups had comparable distributions of coronary artery disease risk factors, including diabetes mellitus, hypertension, and hyperlipidemia, as well as similar medication use profiles (e.g., aspirin, beta-blockers, ACE inhibitors).

In total, 49 stents were placed among the 41 patients. These stents were distributed across the major coronary arteries: 23 stents were deployed in the LAD, 5 in the LCX, and 21 in the RCA.

Patient characteristics stratified by the presence or absence of in-stent restenosis (ISR) at follow-up are presented in Table 2. Among the 41 patients, 10 experienced ISR (ISR ( + )), while 31 had patent stents (ISR (−)). The two groups did not exhibit statistically significant differences in key parameters, including gender distribution, number and location of stented vessels, and cardiovascular risk factors (e.g., diabetes, hypertension, and smoking status).

Although ISR (+) patients tended to have higher rates of hyperlipidemia (80% vs. 45.2%, *p* = 0.075) and a higher proportion of stents in the LCX (20% vs. 9.7%, *p* = 0.580), these differences were not statistically significant due to the small sample size. Additionally, both groups had similar patterns of medication use at baseline, indicating comparable medical management post-PCI.

This analysis highlights that ISR development was not strongly associated with baseline clinical characteristics or stent location, underscoring the potential role of functional myocardial parameters such as TpV and SDTpV in predicting restenosis risk.

### 3.2. Changes in LVEF and Pulse-Wave TDI

The study evaluated the longitudinal changes in LVEF and SDTpV over six months following PCI, with a focus on differences between patients with and without ISR.

Patients were stratified based on ISR status and their presenting condition (AMI or angina pectoris). The TpVc and SDTpVc were TpV and SDTpV corrected by heart rate and were calculated by the value divided by the square root of the ECG R-R interval in milliseconds (Table 3). Among the 41 patients included, 10 patients experienced ISR, while 31 maintained stent patency. The data trends are visually represented in Figure 3, which includes key comparisons between subgroups.

In the overall cohort (Figure 3A,B), patients without ISR demonstrated significant improvements in myocardial function over six months. LVEF increased from 58.8 ± 11.0% at baseline to 66.3 ± 7.7% (*p* < 0.001), and SDTpV decreased from 85.8 ± 44.8 ms to 60.3 ± 31.7 ms (*p* < 0.001), indicating improved myocardial contractility and reduced dyssynchrony. In contrast, patients with ISR showed no significant change in LVEF (65.0 ± 11.6% to 66.7 ± 6.0%, *p* = 0.733) or SDTpV (97.7 ± 53.3 ms to 91.2 ± 52.6 ms, *p* = 0.567). However, mean TpV did not significantly change in the ISR (−) and ISR (+) groups, remaining stable from 249.5 ± 47.9 ms to 249.1 ± 41.4 ms (*p* = 0.966) and from 287.8 ± 54.1 ms to 280.8 ± 44.3 ms (*p* = 0.602), respectively. Mean TpVc, the heart rate-corrected TpV, followed a similar trend, showing no significant differences between baseline and follow-up in either the ISR (−) (271.4 ± 58.6 ms to 270.4 ± 49.2 ms, *p* = 0.910) or ISR (+) (309.5 ± 66.9 ms to 302.2 ± 58.2 ms, *p* = 0.630) groups.

In the AMI subgroup (Figure 3C,D), patients without ISR (*n* = 13) showed marked recovery in LVEF, increasing from 53.1 ± 12.0% to 64.0 ± 10.0% (*p* = 0.005), with a corresponding decrease in SDTpV from 108.7 ± 46.0 ms to 72.4 ± 35.5 ms (*p* = 0.008). Mean TpV remained stable (257.6 ± 48.5 ms to 249.4 ± 38.6 ms, *p* = 0.593). Similarly, in the ISR (+) AMI subgroup (*n* = 6), LVEF did not significantly improve (61.8 ± 13.1% to 66.8 ± 12.3%, *p* = 0.482), and mean TpV remained prolonged (274.2 ± 52.0 ms to 264.4 ± 32.5 ms, *p* = 0.663).

In the angina subgroup (Figure 3G,H), a similar trend was observed. Patients without ISR (*n* = 18) had a significant increase in LVEF from 62.9 ± 8.4% to 67.9 ± 5.3% (*p* = 0.009) and a reduction in SDTpV from 69.3 ± 37.0 ms to 51.6 ± 26.3 ms (*p* = 0.015). However, mean TpV did not significantly change (243.6 ± 47.9 ms to 248.9 ± 44.5 ms, *p* = 0.579). In contrast, ISR (+) angina patients (*n* = 4) exhibited no significant change in LVEF (69.8 ± 7.9% to 66.5 ± 9.6%, *p* = 0.687), while mean TpV remained prolonged (308.1 ± 58.0 ms to 305.2 ± 52.7 ms, *p* = 0.824).

In addition, in both the overall cohort and subgroups, mean peak velocity (Vp) and SD Vp did not differ significantly between ISR (+) and ISR (−) groups, even in patients with AMI or angina presentations. These findings suggest that peak velocity measures were not associated with ISR risk.

### 3.3. TpV as a Predictor of ISR

Mean TpV was analyzed as a key predictor of ISR at six months post-PCI. Logistic regression analysis identified mean TpV as the sole independent predictor of ISR, with an odds ratio of 2.09 (95% CI: 1.004–4.352, *p* = 0.049) (Table 4). ROC curve analysis established a cut-off value of 279.6 ms, yielding a sensitivity of 60% and specificity of 77% (Figure 4). These findings suggest that patients with prolonged TpV are at higher risk of developing ISR.

Figure 5 illustrates the distribution of mean TpV at six months post-PCI in patients with and without ISR. Patients who did not develop ISR had a significantly lower mean TpV of 249.1 ms, whereas those who developed ISR had a higher mean TpV of 280.8 ms. Based on ROC curve analysis, the optimal cut-off value for predicting ISR in the cohort was determined to be 279.6 ms. Patients with mean TpV > 279.6 ms were found to be at an increased risk of developing ISR, highlighting the potential role of TpV as a non-invasive echocardiographic marker for ISR prediction.

These results suggest that while LVEF and SDTpV improved in patients without ISR, mean TpV remained relatively stable in both ISR (−) and ISR (+) groups. This indicates that TpV alone may not directly reflect myocardial recovery post-PCI but rather persistent contraction inefficiency in ISR-prone segments, such as diffuse atherosclerotic vessels. In contrast, mean Vp and SD Vp did not differ significantly between ISR groups, reinforcing the notion that velocity-based parameters alone may not be as predictive of ISR as time-based contractile indices.

## 4. Discussion

The major findings were as follows: (1) a significant decrease in SDTpV that was demonstrated in the 13 AMI and 18 chronic angina patients, without ISR at six months after PCI; (2) A cut-off value of mean TpV ≥ 279 ms at six months after PCI was able to predict an ISR (odds ratio: 2.09, 95% CI 1.004–4.352, *p* = 0.049). These results reflect contractile mechanics dysfunction, assessed by Tissue Doppler Imaging with mean TpV, can noninvasively predict six-month ISR in patients after PCI.

### 4.1. Tissue Doppler Imaging in Myocardial Function

Despite its widespread use in clinical cardiology, LVEF has significant limitations that hinder its ability to fully capture myocardial function. One major challenge is that LVEF is highly dependent on hemodynamic loading conditions, including preload, afterload, and contractility. These variables can fluctuate due to changes in patient status, leading to variability in LVEF that may not accurately reflect the underlying contractile state of the myocardium [8]. In conditions like heart failure with preserved ejection fraction and valvular diseases, LVEF often fails to indicate true myocardial dysfunction, masking disease progression until more severe symptoms develop [9]. Additionally, measurement variability is a concern. LVEF assessment by 2D echocardiography can be influenced by operator expertise, image quality, and chest wall anatomy, contributing to inter- and intra-observer differences. More sensitive parameters, such as TDI, global longitudinal strain, and mechanical dispersion, have emerged as complementary tools. Integrating advanced echocardiographic indices, cardiac MRI, and biomarker analysis is increasingly advocated to overcome the shortcomings of LVEF and improve both diagnostic and prognostic evaluations in CAD and other cardiac conditions.

Time intervals within the cardiac cycle have been the subject of intense studies to understand myocardial mechanics and function [10,11]. The myocardial performance index (MPI) was initially proposed as the Tei Index that studied both left and right ventricles in various disease states [12,13,14]. In AMI populations, the authors showed that MPI acquired by TDI is a simple and reproducible measure that provides independent prognostic information on systolic and diastolic function, incremental to conventional 2DE and Doppler flow velocity parameters in patients with ST segment elevation myocardial infarction treated with PCI [15]. Previous studies reported that diastolic dysfunction develops before systolic contractile dysfunction occurs; therefore, using the diastolic time constant Tau to evaluate myocardial relaxation dysfunction during isovolumic period has been a popular subject of study for early markers of LV function deterioration [16,17]. Mollema et al. first studied 124 patients with AMI receiving primary PCI and observed that the LV dyssynchrony quantified by TDI at baseline was strongly related to the extent of LV dilatation at six-month follow-up [18]. Sequential function evaluation using conventional as well as tissue Doppler echocardiography to evaluate LV systolic function in ischemic cardiomyopathy patients is feasible in evaluation of LV myocardium remodeling after ischemia-reperfusion [18].

Solely relying on LVEF to predict survival after MI can be over-simplistic and prone to errors since LVEF could be normal when extensive regional wall motion abnormalities are compensated by other regions with non-ischemic myocardial hyperkinesia. A higher wall motion score index (WMSI) has been observed as a powerful independent predictor of cardiac events, death, and hospitalization for subsequent heart failure, and patients with a higher WMSI at pre-discharge were reported to have worse outcomes in patients after MI. Moller et al. studied 767 patients with AMI and demonstrated that echocardiographic determined LVEF and WMSI on the first day after admission were powerful predictors of all-cause mortality [19]. However, the interpretation of WMSI depended on the clarity of 2D echocardiographic images and was subjective, semi-quantitative, time-consuming, and required long-term feedback training.

### 4.2. TpV as a Non-Invasive Marker of Myocardial Function

PWTDI is a highly sensitive technique for assessing regional myocardial contraction timing, which is particularly advantageous in patients with poor acoustic windows, such as those with chronic obstructive pulmonary disease (COPD) or morbid obesity, where obtaining clear echocardiographic images is challenging. Unlike conventional 2DE-based assessments, PWTDI provides higher spatio-temporal resolution, allowing for the quantification of longitudinal myocardial motion with good reproducibility [6,15,20,21]. TpV is an echocardiographic parameter derived from PW-TDI that reflects the time interval from the onset of myocardial motion to the peak velocity of myocardial contraction. TpV reflects the velocity and timing of myocardial shortening, making it a load-sensitive but functionally informative metric of contractile performance. Shorter TpV (faster peak contraction) indicates stronger contractility and more efficient myocardial force generation. In contrast, prolonged TpV (delayed contraction peak) suggests contractile dysfunction, ischemic delayed myocardial contraction, or dyssynchrony, where the myocardium takes longer to reach peak contraction.

Regional ischemia-reperfusion injury is known to induce myocardial swelling, increase actin-myosin filament distance, and impair force generation, leading to regional delays in myocardial contraction TpV. These alterations in myocardial mechanics may not be easily detected by visual wall motion analysis but they can be effectively assessed through segmental PWTDI measurements. A previous study by Lin et al. demonstrated that segmental myocardial contraction time-to-peak velocity (TpV ≥ 340 ms) was indicative of significant coronary artery stenosis in patients without visually discernible wall motion abnormalities [3]. This study revealed a regional myocardial contraction delay time-to-peak velocity of ≥ 279.6 ms could identify ISR at six months after PCI reperfusion. This finding suggests that milder but persistent myocardial ischemia post-PCI may lead to progressive myocardial contractile dysfunction, which can be detected using TpV before significant global deterioration in LVEF occurs.

Interestingly, the less significant delay in myocardial contraction velocity (TpV ~279.6 ms) post-PCI may reflect myocardial ischemic dysfunction, whereas a more pronounced delay (TpV ≥ 340 ms) may indicate overt ischemic injury and contractile stunning. A formula integrating these values could be useful for tracking the gradual decline in myocardial contractility associated with chronic ischemic exposure.

Notably, in the 31 patients without ISR and in 19 AMI patients regardless of ISR status, LVEF increased significantly at six months post-PCI, consistent with prior studies [2,22,23]. This improvement suggests that successful revascularization promotes myocardial recovery. However, in patients with ISR, persistent prolongation of mean TpV and increased SDTpV suggest impaired myocardial perfusion recovery, emphasizing the need for closer monitoring and potential secondary interventions.

The combination of delayed contraction timing and increased contractile dyssynchrony detected by PWTDI may serve as an early indicator of ISR-related myocardial dysfunction, guiding targeted follow-up strategies.

### 4.3. Study Limitations

There were some limitations in the present study. First, the small sample size (41 patients), along with further subgrouping, may reduce the statistical power and reliability of the findings. Future studies with larger cohorts are needed to validate these findings. Second, we did not analyze correlations between WMSI and other parameters, such as SD-TpV, mean TpV, or LVEF. Additionally, no direct comparisons were made between PWTDI and other non-invasive imaging modalities (e.g., MRI, nuclear imaging) due to limitations such as high cost, radiation exposure, and inferior temporal resolution. Third, the relatively high ISR rate (39%) observed in the AMI subgroup may reflect the severity of disease in these patients, potentially influencing the outcomes. Fourth, the simultaneous acquisition of segmental TpV across 12 to 16 LV segments was not feasible due to the limitations of 2D echocardiography. 3D echocardiography may offer a solution to this limitation in future studies. Furthermore, while real-time acquisition of TpV and Vp data requires only a few seconds, post-acquisition analysis remains labor-intensive compared to 2D speckle-tracking strain evaluation.

## 5. Conclusions

The present study demonstrated that PWTDI-acquired segmental TpV, along with the calculated mean TpV and SDTpV, may be used for early detection of ischemia-related myocardial dysfunction. Moreover, a longitudinal mean time-to-peak contraction ≥ 279.6 ms of 12 non-apical segments of LV can predict ISR at six months post-PCI.

## Figures and Tables

**Figure 1 diagnostics-15-00579-f001:**
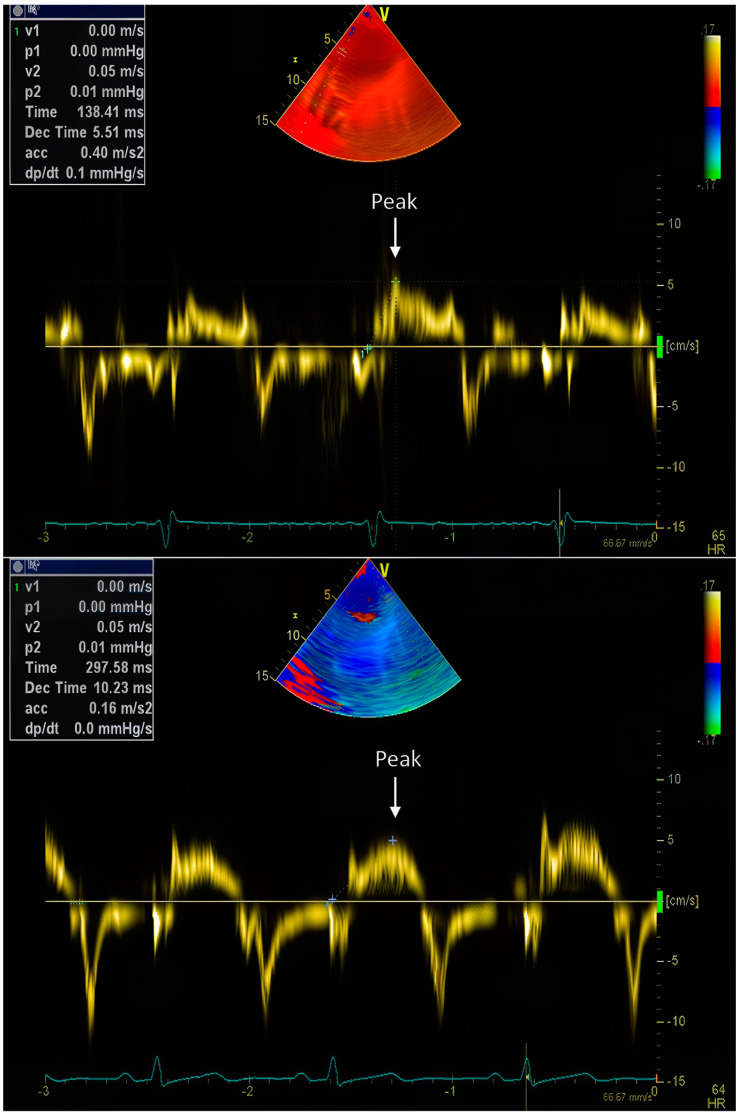
Time-to-peak velocities (TpV) is the measurement of the time period from the ECG R-wave to the point of peak velocity (Vp) toward the apex on tissue velocity integrals. TpV = 138.41 ms (**upper panel**), and TpV = 297.58 ms (**lower panel**).

**Figure 2 diagnostics-15-00579-f002:**
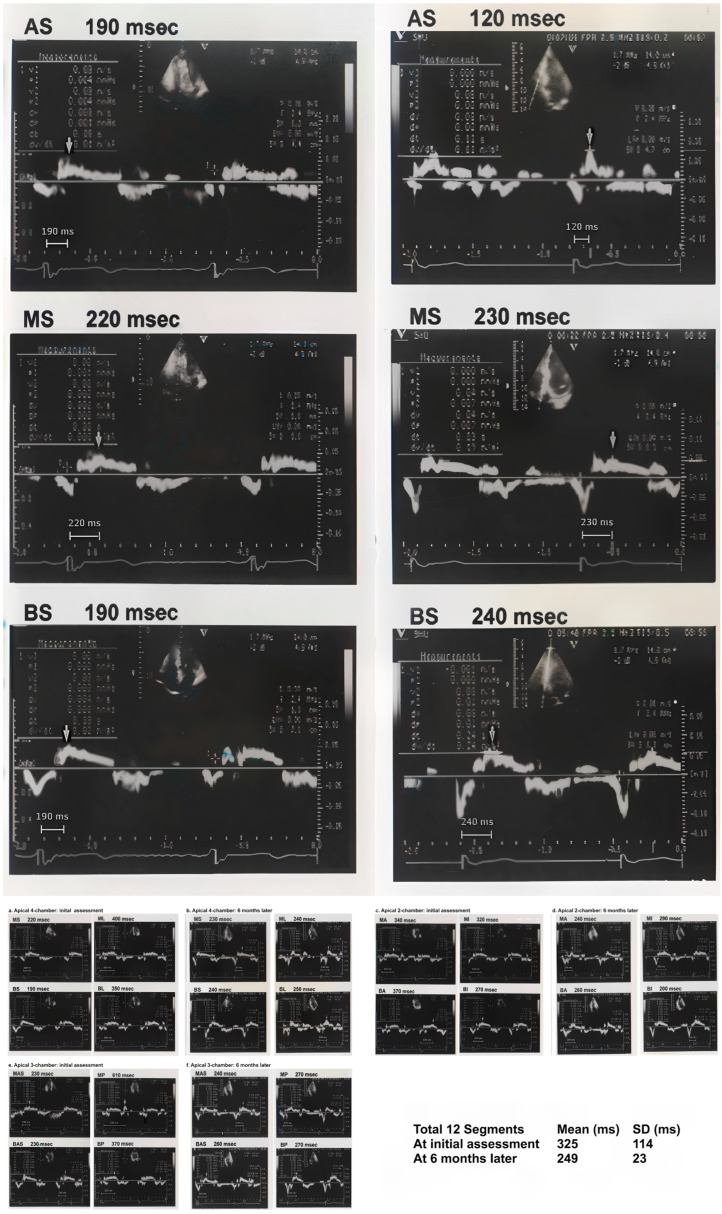
An example of the echocardiographic protocol used in the study. It illustrates the measurement of TpV at each myocardial segment, the averaging process across 12 non-apical LV segments, and the comparison of TpV changes between baseline and six months later. Twelve TpVs were measured at non-apical LV segments in apical four-chamber (A4C), apical two-chamber (A2C), and apical three-chamber (A3C) views to calculate mean ± standard deviation (SD) TpV at the initial assessment and at six months later. Apico LV segments were not measured since myocardial movements in these segments are almost perpendicular to the ultrasound probe and TpV measurements can be unreliable. Mid-septal (MS), baso-septal (BS), mid-lateral (ML), and baso-lateral (BL) were from the A4C view; mid-anterior (MA), baso-anterior (BA), mid-inferior (MI), and baso-inferior (BI) were from A2C view; and mid-anteroseptal (MAS), baso-anteroseptal (BAS), mid-posterior (MP), and baso-posterior (BP) were from the A3C view.

**Figure 3 diagnostics-15-00579-f003:**
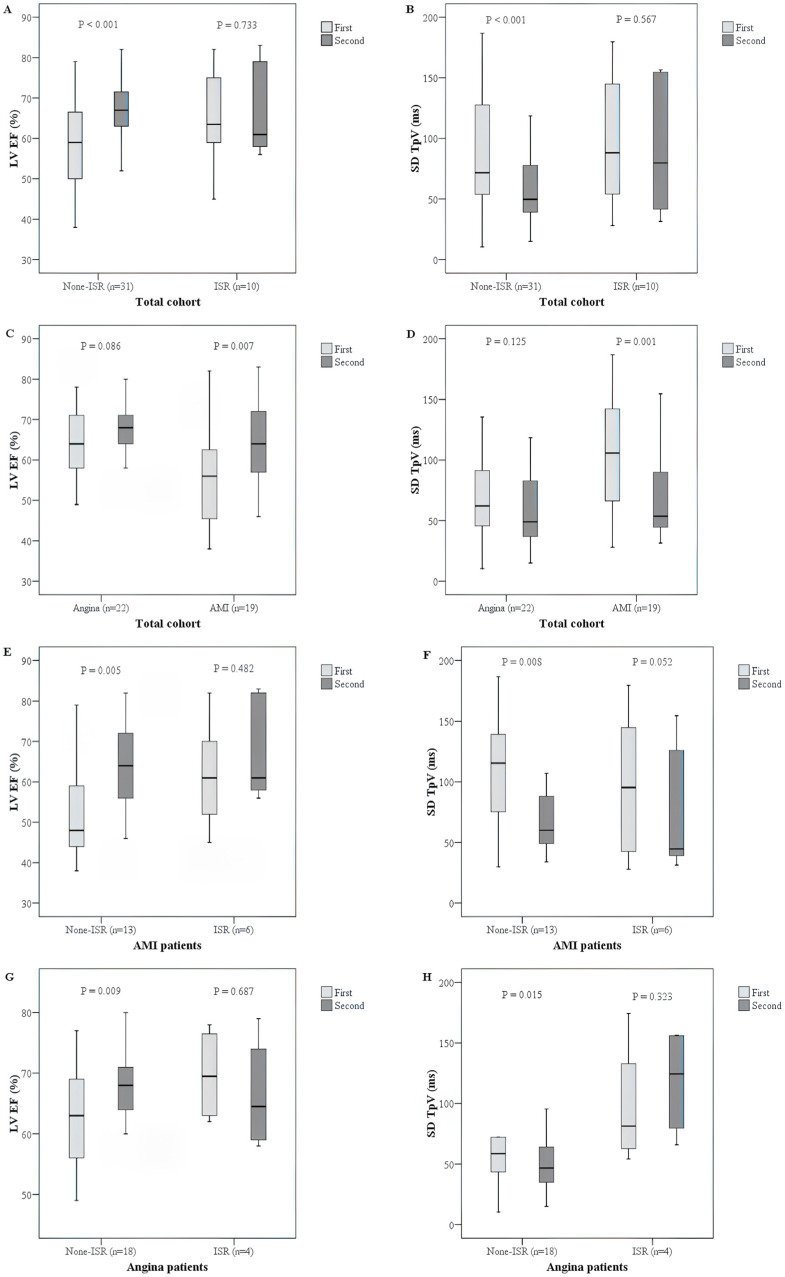
Significant differences of LVEF and SD-TpV in patients without ISR (**A**,**B**). The patients were divided into AMI/angina groups, only the AMI group showed significant change in LVEF and SD-TpV (**C**,**D**). The patients of these AMI and angina groups were further subdivided according to the presence of ISR; only the Non-ISR subgroup in each group had significant change in LVEF or SD-TpV (**E**–**H**).

**Figure 4 diagnostics-15-00579-f004:**
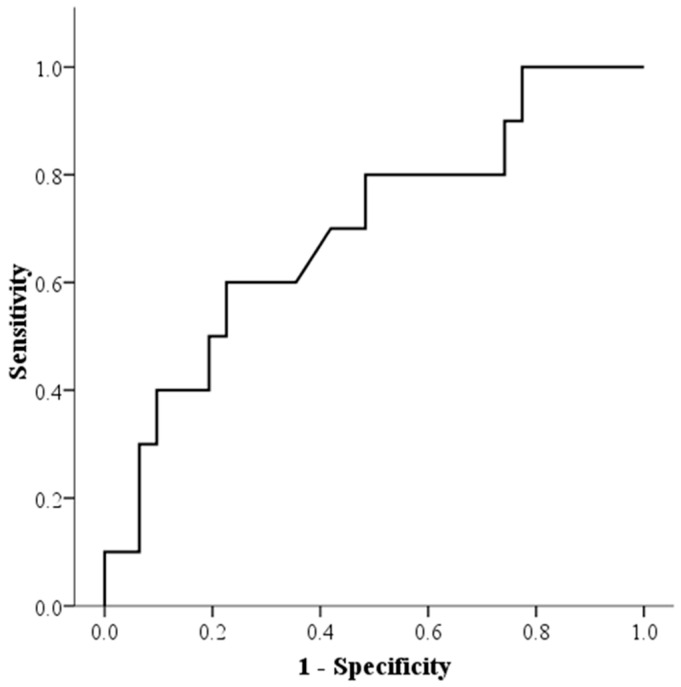
Receiver operating characteristic (ROC) curve for predicting in-stent restenosis (ISR) at six months post-percutaneous coronary intervention (PCI) using mean time-to-peak velocity (TpV). The optimal cut-off value for predicting ISR was 279.6 ms, with a sensitivity of 60% and specificity of 77%. The area under the ROC curve (AUC) was 0.697 (95% CI: 0.506–0.888, *p* = 0.04), indicating a moderate predictive value of mean TpV in ISR detection.

**Figure 5 diagnostics-15-00579-f005:**
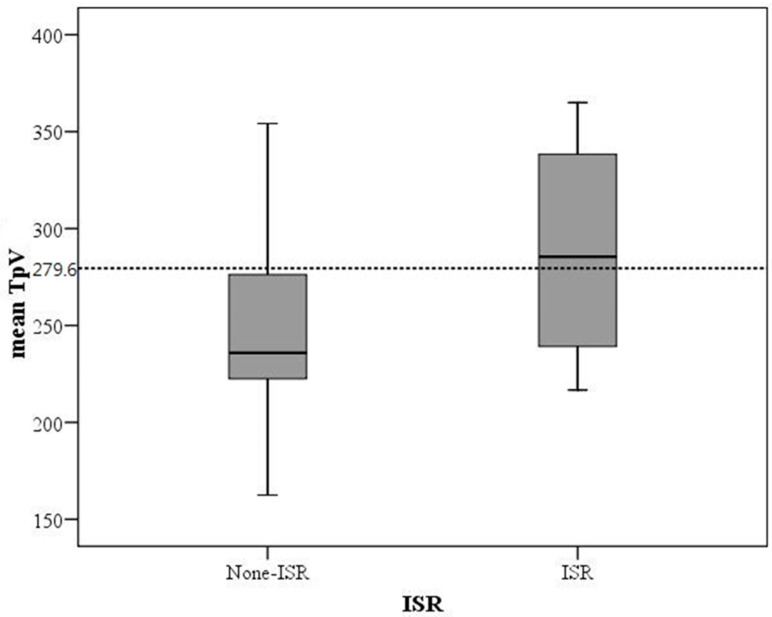
A mean TpV ≥ 279.6 ms at six months after percutaneous coronary intervention was predictive of in-stent restenosis (ISR) (*p* = 0.049).

**Table 1 diagnostics-15-00579-t001:** Baseline Epidemiological, Clinical, Biochemical, and Radiological Characteristics by AMI/Angina Groups.

Parameters	Angina (*N* = 22)	AMI (*N* = 19)	*p*
Categorical variables			
Sex (female), %	4 (18.2)	1 (5.3)	0.350
LAD, %	17 (77.3)	11 (57.9)	0.313
LCX, %	10 (45.5)	6 (31.6)	0.522
RCA, %	12 (54.5)	11 (57.9)	1.000
Number of Vessels, %			0.488
1	10 (45.5)	12 (63.2)	
2	7 (31.8)	5 (26.3)	
3	5 (22.7)	2 (10.5)	
LAD Stent, %	14 (63.6)	9 (47.4)	0.355
LCX Stent, %	4 (18.2)	1 (5.3)	0.350
RCA Stent, %	10 (45.5)	11 (57.9)	0.536
Diabetes mellitus, %	5 (22.7)	7 (36.8)	0.493
Hypertension, %	12 (54.5)	12 (63.2)	0.406
Hyperlipidemia, %	12 (54.5)	10 (52.6)	0.576
Smoking, %	4 (18.2)	10 (52.6)	0.026
ACEI, %	2 (9.1)	7 (36.8)	0.057
ARB, %	2 (9.1)	5 (26.3)	0.219
Beta-blocker, %	19 (86.4)	16 (84.2)	1.000
Calcium-channel blocker, %	5 (22.7)	2 (10.5)	0.419
Aspirin, %	21 (95.5)	17 (89.5)	0.588
Isordil, %	19 (86.4)	15 (78.9)	0.685
Continuous variables			
Age (years) *	64 [52, 75]	60 [53, 67]	0.907
Creatinine (mg/dL) *	1.06 [0.76, 1.21]	1.02 [0.80, 1.59]	0.912
Sodium (mmol/L) *	134 [131, 138]	133 [131, 135]	0.907
Potassium (mmol/L)	4.02 ± 0.27	4.21 ± 0.30	0.042
Hemoglobin (g/dL) *	13.5 [11.6, 15.4]	13.7 [11.3, 15.4]	0.430
White blood count (/μL) *	6320 [5700, 7950]	8020 [6710, 9410]	0.211
Systolic BP (mmHg)	131.4 ± 13.6	142.3 ± 23.1	0.068
Diastolic BP (mmHg)	69.5 ± 8.0	76.9 ± 13.7	0.038
Heart rate (bpm)	68.6 ± 9.7	73.2 ± 12.7	0.209

Continuous variables are expressed as mean ± standard deviation (SD) when normally distributed and as median [interquartile range (IQR)] when not normally distributed. Variables marked with an asterisk (*) indicate non-normally distributed data. ACEI, angiotensin-converting enzyme inhibitor; AMI, acute myocardial infarction; ARB, angiotensin II receptor blocker; BP, blood pressure; LAD, left anterior descending artery; LCX, left circumflex artery; RCA, right coronary artery.

**Table 2 diagnostics-15-00579-t002:** Baseline Epidemiological, Clinical, Biochemical, and Radiological Characteristics by Restenosis Groups.

Parameters	No Restenosis (*N* = 31)	Restenosis (*N* = 10)	*p*
Categorical variables			
Sex (female), %	3 (9.7)	2 (20.0)	0.580
LAD, %	20 (64.5)	8 (80.0)	0.458
LCX, %	11 (35.5)	5 (50.0)	0.472
RCA, %	18 (58.1)	5 (50.0)	0.724
Number of Vessels, %			0.477
1	17 (54.8)	5 (50.0)	
2	10 (32.3)	2 (20.0)	
3	4 (12.9)	3 (30.0)	
LAD Stent, %	17 (54.8)	6 (60.0)	1.000
LCX Stent, %	3 (9.7)	2 (20.0)	0.580
RCA Stent, %	16 (51.6)	5 (50.0)	1.000
Diabetes mellitus, %	7 (22.6)	5 (50.0)	0.124
Hypertension, %	17 (54.8)	7 (70.0)	0.480
Hyperlipidemia, %	14 (45.2)	8 (80.0)	0.075
Smoking, %	11 (35.5)	3 (30.0)	1.000
ACEI, %	6 (19.4)	3 (30.0)	0.662
ARB, %	6 (19.4)	1 (10.0)	0.660
Beta-blocker, %	25 (80.6)	10 (100.0)	0.307
Calcium channel blocker, %	7 (22.6)	0 (0.0)	0.164
Aspirin, %	29 (93.5)	9 (90.0)	1.000
Isordil, %	24 (77.4)	10 (100.0)	0.164
Continuous variables			
Age (years) *	62 [51, 69]	67 [56, 71]	0.834
Creatinine (mg/dL) *	1.25 [0.77, 1.45]	1.19 [0.79, 1.50]	0.768
Sodium (mmol/L) *	134 [132, 136]	133 [131, 138]	0.834
Potassium (mmol/L) *	3.9 [3.7, 4.3]	4.4 [4.1, 4.6]	0.010
Hemoglobin (g/dL)	13.57 ± 2.03	13.61 ± 2.53	0.960
White blood count (/μL) *	6570 [5600, 7900]	8050 [6870, 9150]	0.703
Systolic BP (mmHg)	135.3 ± 20.4	140.1 ± 14.8	0.497
Diastolic BP (mmHg)	72.4 ± 11.6	74.9 ± 11.3	0.497
Heart rate (bpm)	71.1 ± 11.7	69.4 ± 10.0	0.678

Continuous variables are expressed as mean ± standard deviation (SD) when normally distributed and as median [interquartile range (IQR)] when not normally distributed. Variables marked with an asterisk (*) indicate non-normally distributed data.

**Table 3 diagnostics-15-00579-t003:** Comparison of Pulse-wave Tissue Doppler Imaging parameters in AMI/angina Groups.

Groups		LVEF	Mean TpV	SD TpV	Mean TpVc	SD TpVc	Mean Vp	SD Vp
All	1st	60.3 ± 11.3	258.8 ± 51.5	88.7 ± 46.6	280.7 ± 62.1	96.8 ± 52.4	6.27 ± 0.97	1.39 ± 0.41
(*N* = 41)	2nd	66.4 ± 8.4	256.8 ± 43.8	67.9 ± 39.4	278.1 ± 52.6	73.5 ± 42.8	6.25 ± 0.93	1.45 ± 0.38
	*p*	0.001	0.777	0.001	0.739	0.001	0.857	0.290
ISR(+)	1st	65.0 ± 11.6	287.8 ± 54.1	97.7 ± 53.3	309.5 ± 66.9	106.7 ± 61.8	5.80 ± 0.74	1.27 ± 0.30
(*N* = 10)	2nd	66.7 ± 10.7	280.8 ± 44.3	91.2 ± 52.6	302.2 ± 58.2	99.4 ± 60.3	5.78 ± 0.68	1.27 ± 0.31
	*p*	0.733	0.602	0.567	0.630	0.561	0.903	0.981
ISR(−)	1st	58.8 ± 11.0	249.5 ± 47.9	85.8 ± 44.8	271.4 ± 58.6	93.6 ± 49.7	6.43 ± 1.00	1.43 ± 0.44
(*N* = 31)	2nd	66.3 ± 7.7	249.1 ± 41.4	60.3 ± 31.7	270.4 ± 49.2	65.1 ± 32.5	6.40 ± 0.96	1.51 ± 0.39
	*p*	<0.001	0.966	<0.001	0.910	<0.001	0.887	0.239
AMI	1st	55.8 ± 12.7	262.9 ± 48.8	105.2 ± 48.8	289.3 ± 60.3	116.3 ± 56.2	6.29 ± 0.81	1.31 ± 0.44
(*N* = 19)	2nd	64.9 ± 10.5	254.2 ± 36.6	72.7 ± 40.2	279.5 ± 45.5	79.9 ± 44.1	6.48 ± 0.91	1.43 ± 0.45
	*p*	0.007	0.474	0.001	0.459	0.001	0.338	0.236
Angina	1st	64.2 ± 8.5	255.3 ± 54.7	74.5 ± 40.5	273.2 ± 64.1	79.9 ± 43.3	6.26 ± 1.12	1.46 ± 0.38
(*N* = 22)	2nd	67.7 ± 6.0	259.2 ± 49.9	63.7 ± 39.2	277.0 ± 59.2	67.9 ± 41.8	6.05 ± 0.92	1.47 ± 0.32
	*p*	0.086	0.634	0.125	0.665	0.120	0.205	0.857
AMI	1st	61.8 ± 13.1	274.2 ± 52.0	97.6 ± 58.4	302.2 ± 72.9	110.5 ± 71.9	6.08 ± 0.77	1.23 ± 0.31
ISR(+)	2nd	66.8 ± 12.3	264.4 ± 32.5	73.4 ± 52.8	291.7 ± 57.2	83.5 ± 65.2	6.02 ± 0.62	1.22 ± 0.31
(*N* = 6)	*p*	0.482	0.663	0.052	0.680	0.049	0.821	0.928
AMI	1st	53.1 ± 12.0	257.6 ± 48.5	108.7 ± 46.0	283.4 ± 55.8	119.0 ± 50.6	6.39 ± 0.83	1.34 ± 0.49
ISR(−)	2nd	64.0 ± 10.0	249.4 ± 38.6	72.4 ± 35.5	273.8 ± 40.3	78.2 ± 33.7	6.70 ± 0.96	1.52 ± 0.49
(*N* = 13)	*p*	0.005	0.593	0.008	0.565	0.009	0.253	0.164
Angina	1st	69.8 ± 7.9	308.1 ± 58.0	97.8 ± 53.2	320.3 ± 65.5	101.0 ± 52.5	5.38 ± 0.51	1.34 ± 0.30
ISR(+)	2nd	66.5 ± 9.6	305.2 ± 52.7	117.8 ± 45.4	317.9 ± 64.5	123.1 ± 50.5	5.41 ± 0.67	1.35 ± 0.33
(*N* = 4)	*p*	0.687	0.824	0.323	0.853	0.323	0.935	0.922
Angina	1st	62.9 ± 8.4	243.6 ± 47.9	69.3 ± 37.0	262.7 ± 60.7	75.2 ± 41.3	6.45 ± 1.13	1.49 ± 0.40
ISR(−)	2nd	67.9 ± 5.3	248.9 ± 44.5	51.6 ± 26.3	267.9 ± 55.8	55.7 ± 28.9	6.19 ± 0.93	1.50 ± 0.32
(*N* = 18)	*p*	0.009	0.579	0.015	0.622	0.013	0.164	0.883

ISR, in-stent restenosis; LVEF, left ventricular ejection fraction; SD TpV, standard deviation of time-to-peak velocities; SD TpVc, corrected standard deviation of time-to-peak velocities; SD Vp, standard deviation of peak velocity; TpV, time-to-peak velocities; TpVc, corrected time-to-peak velocities; Vp, peak velocity.

**Table 4 diagnostics-15-00579-t004:** Adjusted Odds Ratio (95% CI) of the Potential Predictor of Restenosis using TpVs, LVEF, and Vps (Forward LR multivariate logistic regression).

Predictors	B	*p*	OR	95% CI for OR
Mean TpV	0.737	0.049	2.090	1.004–4.352

Forward LR multivariate logistic regression analysis indicated that mean time-to-peak velocities (TpV) were independently associated with an increased probability of restenosis (95% CI of OR = 1.004–4.352).

## Data Availability

The data presented in this study are available on request from the corresponding author. The data are not publicly available due to ethical restrictions.

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
