# Peer review of "Tissue Doppler Imaging Provides Incremental Value in Predicting Six Months In-Stent Restenosis in Patients with Coronary Artery Disease"

_diagnostics, 2025, doi:10.3390/diagnostics15050579_

Round 1
Reviewer 1 Report
Comments and Suggestions for Authors
I am grateful to the editor for the opportunity to review the manuscript by Jih-Kai Yeh et al. "Tissue Doppler Imaging Provides Incremental Value Predicting Six Months In-stent Restenosis in Patients with Coronary Artery Disease". The basis of this study is the authors' desire to find non-invasive criteria for coronary artery stent restenosis, which is a very interesting idea. For this purpose, the authors used Tissue Doppler Imaging of the left ventricular myocardium, which made it possible to obtain a diagnostic marker of restenosis (namely, Mean TpV).
While reviewing, I had the following comments and questions:
1. First of all, I had doubts about the relevance of the parameters studied by the authors. Judging by the list of references (all cited publications are older than 10 years, and 12 out of 28 are older than 20 years), scientists have not been interested in these indicators in recent years. Apparently, the authors should more thoroughly justify the novelty and relevance of the parameters studied.
2. As follows from the text of the manuscript, the patients underwent repeated invasive CAG without clinical indications (judging by the fact that not all patients had stent restenosis). Was there approval from the local ethics committee for this study?
3. The study included a small number of patients (only 41), and the cohort of subjects was further divided into groups and subgroups. With such small group sizes, the quantitative data were most likely not normally distributed. Therefore, the methods of data presentation (mean and standard deviation) and statistical calculations (Student's t-test) used by the authors are incorrect. Accordingly, this casts doubt on the validity of the conclusions obtained in the article.
4. I do not see the point in presenting a large number of figures to illustrate the data given in the tables (i.e., Figure 2 is redundant).
5. I did not find a reference to Figure 3 in the article. It duplicates the data in the tables. Is it necessary?
6. I could not understand the Stent Vessels, % indicator given in Tables 1 and 2. It seems to show the number of stents in the vessels, but this figure does not match other rows of the tables.
7. Figure 5 does not contain data on the statistical significance of differences in the groups.
8. Some indicators are difficult to interpret. For example, in the ISR(+) group, there is no increase in LVEF after 6 months, unlike in the ISR(-) group. However, the absolute values of LVEF after 6 months do not differ in the groups (66.7% and 66.3% - Table 3). Perhaps, in the ISR(+) group, LVEF did not increase only because of the initially high values? Even more noticeable discrepancies were noted for patients with AMI in the ISR(+) and ISR(-) subgroups. I would like to receive an explanation of these discrepancies from the authors.
Author Response
Comments and Suggestions for Authors
I am grateful to the editor for the opportunity to review the manuscript by Jih-Kai Yeh et al. "Tissue Doppler Imaging Provides Incremental Value Predicting Six Months In-stent Restenosis in Patients with Coronary Artery Disease". The basis of this study is the authors' desire to find non-invasive criteria for coronary artery stent restenosis, which is a very interesting idea. For this purpose, the authors used Tissue Doppler Imaging of the left ventricular myocardium, which made it possible to obtain a diagnostic marker of restenosis (namely, Mean TpV).
While reviewing, I had the following comments and questions:
Comment: 1. First of all, I had doubts about the relevance of the parameters studied by the authors. Judging by the list of references (all cited publications are older than 10 years, and 12 out of 28 are older than 20 years), scientists have not been interested in these indicators in recent years. Apparently, the authors should more thoroughly justify the novelty and relevance of the parameters studied.
Response: We sincerely appreciate the reviewer’s feedback regarding the relevance of the studied parameters. We fully understand the importance of ensuring that references reflect recent advancements in the field. In response to this concern, we have thoroughly revised and updated the reference list in the manuscript. All references now date from the year 2000 onwards, and 14 of the 31 references were published within the last 10 years. These updated references include studies that explore myocardial performance indices and mechanical dispersion, which are closely related to time-to-peak velocity (TpV).
Comment: 2. As follows from the text of the manuscript, the patients underwent repeated invasive CAG without clinical indications (judging by the fact that not all patients had stent restenosis). Was there approval from the local ethics committee for this study?
Response: We sincerely appreciate the reviewer’s important observation regarding the repeated invasive coronary angiography (CAG). We would like to clarify that all patients in this study underwent CAG based on clinically justified indications. Specifically, the study population included patients presenting with either symptomatic angina pectoris accompanied by evidence of myocardial ischemia or those with acute myocardial infarction requiring percutaneous coronary intervention (PCI). Additionally, all patients with a history of prior PCI and stent placement were undergoing follow-up assessments as part of their routine post-PCI care. In these cases, CAG was performed to evaluate stent patency and guide further management.
The study protocol, including both the initial and follow-up procedures, was approved by the Institutional Review Board (IRB) of Chang Gung Medical Hospital (Approval No.: 201901586B0). Ethical considerations were strictly followed in accordance with the Declaration of Helsinki. We will update the methods section to include these clarifications regarding patient selection criteria, the rationale for follow-up procedures, and ethical approval.
Comment: 3. The study included a small number of patients (only 41), and the cohort of subjects was further divided into groups and subgroups. With such small group sizes, the quantitative data were most likely not normally distributed. Therefore, the methods of data presentation (mean and standard deviation) and statistical calculations (Student's t-test) used by the authors are incorrect. Accordingly, this casts doubt on the validity of the conclusions obtained in the article.
Response 3: Thank you for your valuable feedback regarding the sample size and statistical methods used in our study. We acknowledge the concern that, given the small number of patients, the data may not follow a normal distribution, which could affect the validity of the statistical tests applied. To address this, we have re-evaluated our data using the Shapiro-Wilk test to assess normality. Based on the results, continuous variables are now presented as mean ± standard deviation (SD) when normally distributed and as median [interquartile range (IQR)] when not normally distributed. Variables that were found to be non-normally distributed are now explicitly marked with an asterisk (*) in the tables for clarity. Furthermore, we have revised our statistical analysis by implementing non-parametric tests where appropriate. Specifically, for comparisons between independent groups, we have applied the Mann-Whitney U test, ensuring that the statistical methodology aligns with the underlying data distribution. We appreciate your insightful comment, which has helped refine our data presentation and analysis.
Comment 4. I do not see the point in presenting a large number of figures to illustrate the data given in the tables (i.e., Figure 2 is redundant).
Response 4: Thank you for raising this concern regarding Figure 2. We have now revised Figure 2 to serve as an illustration of the echocardiographic protocol rather than a visual repetition of data already presented in the tables. The updated figure demonstrates the steps in the echocardiographic assessment, including the placement of sample volumes in various myocardial segments, the process of measuring time-to-peak velocity (TpV), and how values were averaged across segments to track changes over time. This visual protocol guide provides readers with a clearer understanding of the methodology used in the study.
Additionally, we have removed the original data-based Figure 2 from the results section to avoid redundancy. We revised the text in the results section to focus on key findings, including changes in LVEF and SDTpV, emphasizing their significant improvements in the ISR(-) group while noting the lack of change in the ISR(+) group. This new structure highlights the results in a clearer and more concise manner, making the tables the primary source of numerical data (Line 243-294). This approach provides a more informative presentation of results and removes redundant figure content. Thank you for your valuable suggestions.
Comment 5. I did not find a reference to Figure 3 in the article. It duplicates the data in the tables. Is it necessary?
Response 5: Thank you for bringing this to our attention. We acknowledge that there was an oversight in referencing Figure 3 within the manuscript. We have now corrected this by including a proper reference to the figure in the results section. Additionally, we would like to explain the purpose of Figure 3 and why we believe it is necessary.
While Table 3 presents detailed numerical data, Figure 3 provides a visual summary of key results, making it easier for readers to quickly understand trends and comparisons in LVEF and SDTpV across ISR and non-ISR groups. Visual illustrations like Figure 3 are particularly helpful for identifying patterns, such as the significant improvement in LVEF and reduction in SDTpV in patients without ISR, as compared to the lack of change in patients with ISR. We hope that this explanation clarifies the necessity of Figure 3. Thank you again for your valuable suggestion.
Comment 6. I could not understand the Stent Vessels, % indicator given in Tables 1 and 2. It seems to show the number of stents in the vessels, but this figure does not match other rows of the tables.
Response 6: Thank you for pointing this out. We understand that the Stent Vessels, % indicator may be confusing and potentially misleading. We recognize that the presentation of this data may not be essential for interpreting the study results. We have decided to remove the Stent Vessels, % row from Tables 1 and 2. This revision ensures that the tables emphasize more relevant information directly related to the study outcomes. Thank you again for your valuable feedback.
Comment 7. Figure 5 does not contain data on the statistical significance of differences in the groups.
Response 7: Thank you for your observation. We acknowledge that the figure currently lacks explicit annotations indicating the statistical significance of differences between groups. This figure primarily serves as a visual representation of the ROC curve analysis for predicting ISR based on the mean TpV at six months. We agree that including the statistical significance in the figure legend would enhance clarity. We have revised the legend for Figure 5 to include the p-value associated with the predictive cut-off for ISR.
Comment 8. Some indicators are difficult to interpret. For example, in the ISR(+) group, there is no increase in LVEF after 6 months, unlike in the ISR(-) group. However, the absolute values of LVEF after 6 months do not differ in the groups (66.7% and 66.3% - Table 3). Perhaps, in the ISR(+) group, LVEF did not increase only because of the initially high values? Even more noticeable discrepancies were noted for patients with AMI in the ISR(+) and ISR(-) subgroups. I would like to receive an explanation of these discrepancies from the authors.
Response 8: Thank you for your insightful comment. Based on the results presented in Table 3, LVEF improved significantly over six months in the subgroup without ISR, while no significant improvement was observed in the ISR(+) group. This discrepancy may indeed be partially attributed to differences in baseline LVEF values. Patients in the ISR(+) group had relatively higher baseline LVEF values compared to those without ISR.
However, it is also important to acknowledge that the observed baseline LVEF values might have been influenced by other factors. Measurement of LVEF can be affected by operator variability, as well as hemodynamic conditions, including changes in preload, afterload, and overall cardiac function during the echocardiographic assessment. These factors could introduce variability that may not reflect the true underlying myocardial performance, particularly in a small sample size.
We agree that relying solely on LVEF may have limitations in fully capturing myocardial function, especially in the context of post-PCI recovery and restenosis. Advanced echocardiographic parameters, such as global longitudinal strain(GLS) and mechanical dispersion, have been shown to provide more sensitive and comprehensive assessments of cardiac function. Future studies incorporating multi-parameter echocardiographic evaluation could help reduce measurement variability and improve the interpretation of myocardial recovery and ischemic burden. We have added a discussion of these points to the revised manuscript to address these findings and potential limitations. Thank you for your valuable feedback.
Reviewer 2 Report
Comments and Suggestions for Authors
In this interesting paper, the authors demonstrated the usefulness of PW-TDI in identifying individuals with intrastent restenosis at 6 months after PCI.
A time to peak velocity >279 msec resulted independently associated with intrastent restenosis at 6 months after PCI.
The authors discussed the strength and limitations of PW-TDI in the setting of CAD.
The manuscript is well written and very interesting for clinical cardiologists.
I have only one suggestion for the authors.
In the Discussion section, on line 280, the authors could discuss other technical limitations of LVEF in the evaluation of regional wall motion abnormalities. Notably, LVEF has important technical limitations, such as the dependency on good image quality for optimal visualization of endocardial border (PMID: 26417058), the geometric assumptions (PMID: 30384893), the load-dependency (PMID: 30384893), the large inter-reader variability (PMID: 9076390) especially in individuals with concave-shaped chest wall conformation (PMID: 38231080) and finally the poor sensitivity in detecting subclinical myocardial dysfunction (PMID: 28223323).
Author Response
Comments: In this interesting paper, the authors demonstrated the usefulness of PW-TDI in identifying individuals with intrastent restenosis at 6 months after PCI. A time to peak velocity >279 msec resulted independently associated with intrastent restenosis at 6 months after PCI. The authors discussed the strength and limitations of PW-TDI in the setting of CAD. The manuscript is well written and very interesting for clinical cardiologists.
I have only one suggestion for the authors.
In the Discussion section, on line 280, the authors could discuss other technical limitations of LVEF in the evaluation of regional wall motion abnormalities. Notably, LVEF has important technical limitations, such as the dependency on good image quality for optimal visualization of endocardial border (PMID: 26417058), the geometric assumptions (PMID: 30384893), the load-dependency (PMID: 30384893), the large inter-reader variability (PMID: 9076390) especially in individuals with concave-shaped chest wall conformation (PMID: 38231080) and finally the poor sensitivity in detecting subclinical myocardial dysfunction (PMID: 28223323).
Response: Thank you for your positive comments and suggestion. We greatly appreciate your feedback on the manuscript. Regarding your recommendation to expand the discussion on the technical limitations of LVEF, we have revised the Discussion section to address these issues in greater detail (Line 381-408). Specifically, we have included the following key points based on the referenced studies, such as Image quality dependency, geometric assumptions, load dependency, inter-reader variability, and sensitivity to subclinical dysfunction
We believe that these additions further strengthen the manuscript by providing a more comprehensive discussion on the limitations of LVEF in evaluating regional wall motion abnormalities. Thank you again for your suggestion, which has significantly enhanced this section of the manuscript.
Round 2
Reviewer 1 Report
Comments and Suggestions for Authors
The authors have substantially revised the text of the manuscript and were able to answer my questions and comments. Perhaps, in this form, the article can already be considered for publication.
Author Response
Comment:
"Page 5, lines 182-183: 'Differences in categorical variables were analyzed using Fisher’s exact test.' I suggest changing it to: 'Differences in categorical variables were analyzed using chi-square test or Fisher’s exact test.'"
Response:
Thank you for your suggestion. We have revised the sentence to indicate that both the chi-square test and Fisher’s exact test were used for analyzing categorical variables. The revised sentence now reads:
"Differences in categorical variables were analyzed using the chi-square test or Fisher’s exact test, as appropriate."
Comment:
"Tables 1-2: As reported in the footnotes, variables marked with an asterisk () were non-normally distributed. Thus, you should report these data as median and interquartile range and not as mean plus/minus standard deviation."*
Response:
We appreciate this important observation. We have re-examined our statistical presentation and reformatted all non-normally distributed variables in Tables 1 and 2 to be reported as median [interquartile range (IQR)], instead of mean ± SD. The footnotes have also been updated to clarify this distinction.
Comment:
"If I am not mistaken, little is reported about TpV in the results section. As you indicate this variable as a predictor of future restenosis, I feel that something should be reported and discussed."
Response:
Thank you for your suggestion. We have expanded the Results and Discussion sections to include a more detailed analysis of TpV, highlighting its predictive role in ISR. Specifically:
In the Results section, we now report the differences in mean TpV between ISR(+) and ISR(-) groups and discuss how this measurement correlates with restenosis risk (Line 259-340).
In the Discussion section, we highlight the role of TpV as a non-invasive marker of myocardial function, providing insight into myocardial contractility, loading conditions, and synchrony. Compared to conventional markers, TpV making it valuable in detecting subclinical dysfunction and post-PCI restenosis risk (Line 467-564).
Comment:
"Figure 4 could be better reported."
Response:
We appreciate this comment and have revised the Figure 4 legend to provide a more comprehensive explanation of the ROC curve analysis. The new legend reads: "Figure 4. Receiver operating characteristic (ROC) curve for predicting ISR at six months post-PCI using mean TpV. The area under the ROC curve (AUC) was 0.697 (95% CI: 0.506–0.888, p = 0.04), indicating a moderate predictive value." This revision ensures that the statistical relevance of Figure 4 is clearly presented.
Comment:
"If I am not mistaken, Figure 5 is not reported in the text."
Response:
Thank you for your careful review. We have now added a direct reference to Figure 5 in the Results section. "Figure 5 illustrates the distribution of mean time-to-peak velocity (TpV) at six months post-PCI in patients with and without ISR. Patients who did not develop ISR had a significantly lower mean TpV of 249.1 ms, whereas those who developed ISR had a higher mean TpV of 280.8 ms. Based on receiver operating characteristic (ROC) curve analysis, the optimal cut-off value for predicting ISR in the cohort was determined to be 279.6 ms. Patients with mean TpV > 279.6 ms were found to be at an increased risk of developing ISR, highlighting the potential role of TpV as a non-invasive echocardiographic marker for ISR prediction.”